# The degree of polymerization and sulfation patterns in heparan sulfate are critical determinants of cytomegalovirus entry into host cells

**Dipanwita Mitra**[1ᵒ], **Mohammad H. Hasan**[1ᵒ], **John T. Bates**[1,2], **Michael A. Bierdeman**[2], **Dallas R. Ederer**[1], **Rinkuben C. Parmar**[1], **Lauren A. Fassero**[1], **Quntao Liang**[3,4], **Hong Qiu**[5], **Vaibhav Tiwari**[6], **Fuming Zhang**[7], **Robert J. Linhardt**[7], **Joshua S. Sharp**[3], **Lianchun Wang**[8], **Ritesh Tandon**[1,2,3]*

1 Department of Microbiology and Immunology, University of Mississippi Medical Center, Jackson, Mississippi, United States of America, 2 Department of Medicine, University of Mississippi Medical Center, Jackson, Mississippi, United States of America, 3 Biomolecular Sciences, School of Pharmacy, University of Mississippi, Oxford, Mississippi, United States of America, 4 College of Biological Science and Engineering, University of Fuzhou, Fujian, China, 5 Complex Carbohydrate Research Center, University of Georgia, Athens, Georgia, United States of America, 6 Department of Microbiology and Immunology, Midwestern University, Downers Grove, Illinois, United States of America, 7 Center for Biotechnology and Interdisciplinary Studies, Rensselaer Polytechnic Institute, Troy, New York, United States of America, 8 Department of Molecular Pharmacology and Physiology, University of South Florida, Tampa, Florida, United States of America

ᵒ These authors contributed equally to this work.

* rtandon@umc.edu

**Data Availability Statement:** All data are in the manuscript and/or Supporting information files.

## Abstract

Several enveloped viruses, including herpesviruses attach to host cells by initially interacting with cell surface heparan sulfate (HS) proteoglycans followed by specific coreceptor engagement which culminates in virus-host membrane fusion and virus entry. Interfering with HS-herpesvirus interactions has long been known to result in significant reduction in virus infectivity indicating that HS play important roles in initiating virus entry. In this study, we provide a series of evidence to prove that specific sulfations as well as the degree of polymerization (*dp*) of HS govern human cytomegalovirus (CMV) binding and infection. First, purified CMV extracellular virions preferentially bind to sulfated longer chain HS on a glycoarray compared to a variety of unsulfated glycosaminoglycans including unsulfated shorter chain HS. Second, the fraction of glycosaminoglycans (GAG) displaying higher *dp* and sulfation has a larger impact on CMV titers compared to other fractions. Third, cell lines deficient in specific glucosaminyl sulfotransferases produce significantly reduced CMV titers compared to wild-type cells and virus entry is compromised in these mutant cells. Finally, purified glycoprotein B shows strong binding to heparin, and desulfated heparin analogs compete poorly with heparin for gB binding. Taken together, these results highlight the significance of HS chain length and sulfation patterns in CMV attachment and infectivity.

**Funding:** The research was supported by NASA (Award #80NSSC19K1603, PI: RT) and NIH (Award 5U01CA225784, PI: LW). QL, JSS and LW acknowledge funding from the National Institute of General Medical Sciences through the Research Resource for Integrated Glycotechnology (P41GM103390). RL and FZ were supported by NIH for T200 Biacore instrument grant (S10OD0285232). The funders had no role in study design, data collection and analysis, decision to publish, or preparation of the manuscript.

**Competing interests:** The authors have declared that no competing interests exist.

## Author summary

Heparan sulfate (HS) is a linear polysaccharide (sugar) found in all animal tissues. It binds to a variety of protein ligands, including cytokines, chemokines, growth factors and morphogens and regulates a wide range of biological activities, including developmental processes, angiogenesis, blood coagulation, and tumor metastasis. The molecular diversity in HS chains generates unique binding sites for specific ligands and can offer preferential binding for a specific virus over other viruses or cellular ligands. In the current study human cytomegalovirus (CMV) was found to bind preferentially to uniquely sulfated and polymerized HS using its specific surface glycoprotein (gB). The HS mimics designed to bind gB inhibited CMV infection. The results were corroborated by parallel studies in mutant mouse cells as well as using protein-polysaccharide binding assays. Combined together, the data suggests that CMV preferentially attaches to uniquely modified HS and thus this virus-host interaction is amenable to targeting by specifically designed HS mimics.

## Introduction

The heparan sulfate (HS) proteoglycans are present on most cell types and function as cellular attachment receptors for medically important viruses, including human immunodeficiency virus (HIV), hepatitis-C virus (HCV), human papillomavirus (HPV), Dengue virus (DENV) and the recently emerged SARS-CoV-2 [1–6]. In addition, virtually all human herpesviruses, with the possible exception of Epstein Barr virus, use HS as an initial co-receptor for entry [7]. The interaction between cell surface HS and virus envelope is the initial event in the complex process of virus entry. A successful virus entry involves downstream co-receptor interactions ultimately leading to fusion between the virus envelope and the cell membrane [8].

Herpesviruses and several other enveloped viruses enter the host cells using two distinct pathways: 1) A pH-independent pathway which involves the fusion of the virus envelope with the plasma membrane; and 2) A pH-dependent pathway that involves endocytosis of the virus particle [9]. In cells, where binding of virus to cell surface receptors induces endocytosis, the usual consequence is the acidification of the endosome, which ultimately triggers fusion between the virus envelope and endosomal membrane [7]. Interestingly, human cytomegalovirus (HCMV) entry mainly follows direct fusion at the cell surface in fibroblasts, while entry into other relevant cell types, such as endothelial cells, follows an endocytic route [10,11]. Different virus glycoprotein complexes are involved in each case; however, HS functions as the primary attachment receptor. Moreover, the presence of HS receptors are well documented in endosomal membranes and HS receptors likely play roles in intracellular virus trafficking [12–15].

The herpesvirus envelope is a lipid bilayer derived from host cell membranes in which most cellular proteins have been displaced by viral membrane proteins. For HCMV, at least twenty three different viral glycoproteins have been found to be associated with purified virion preparations [16]. For most herpesviruses, the conserved glycoprotein B (gB) is required for virus entry and it binds to cell surface molecules, including HS, which is present not only as a constituent of cell surface proteoglycans but also as a component of the extracellular matrix and basement membranes in organized tissues [7,17]. HCMV gB binds to HS resulting in virus attachment [18] similar to its counterparts in herpes simplex virus (HSV)-1 [17,19] and varicella-zoster virus (VZV) [20]. Subsequently, HCMV gB binds to cellular protein receptors

such as EGFR [21], PDGFα [22], and integrins [23,24], which culminates in virus entry. Treatment of cells with soluble form of gB inhibits HCMV entry [25]. HCMV binding and infection are reduced by soluble heparin and HS, as well as in cells treated with heparinases or those unable to produce HS [26]. A better structural understanding of these inhibitions will pave the way to design effective antivirals that are highly specific as well as effective.

The synthesis of HS is a complex process involving multiple specialized enzymes and is initiated from a tetrasaccharide (GlcA-Gal-Gal-Xyl) that is attached to the core protein [27–29]. HS polymerase is responsible for building the polysaccharide backbone with a repeating unit of -GlcA-GlcNAc-. The backbone is then modified by N-deacetylase/N-sulfotransferase (NDST) responsible for N-deacetylation and N-sulfation of selected glucosamine residues, C5-epimerase responsible for epimerization of selected glucuronic moieties to iduronic acid, 2-O-sulfotransferase (Hs2st; 2-O-ST) responsible for 2-O-sulfation of selected iduronic acid residues, 6-O-sulfotransferase (Hs6st; 6-O-ST) for 6-O-sulfation and finally (but rarely) 3-O-sulfotransferases (Hs3st; 3-O-ST) responsible for 3-O-sulfation [30,31]. The substrate specificities of these biosynthetic enzymes dictate the structures of HS products, including sulfation levels, the contents of L-iduronic acid (IdoA) units and the size of the polysaccharides [30]. The location of the sulfo groups and IdoA in turn play a crucial role in determining the binding and functions of HS.

In the current study, we investigated the impact of specific sulfations as well as degree of polymerization in terms of numbers of monosaccharide units (*dp*) in HS chain on both human and mouse CMV infection and binding. Purified CMV extracellular virions preferentially bound strongly to the longer sulfated HS chains but not to the shorter unsulfated HS chains on a glycoarray. Glycosaminoglycans of different *dp* were derivatized from enoxaparin (ES, a low molecular weight heparin) and tested for their ability to inhibit CMV infection in cell culture. The results show that longer glycan chains are more efficient at reducing CMV titers in cells compared to shorter chain glycans. Also, the cell lines defective in expression of various sulfotransferases showed significantly reduced CMV entry and replication. Finally, purified glycoprotein B showed strong binding to heparin, and desulfated heparin analogs competed poorly with heparin for gB binding. Overall, these results indicate that CMV binding to cell surface glycans is dependent on branch length and sulfation pattern of HS.

## Materials and methods

### Preparation of glycosaminoglycans (GAGs) oligosaccharides

Glycosaminoglycans of different *dp* were fractionated from enoxaparin by Bio-Gel P-10 chromatography as previously described [32]. Briefly, 15 mg/mL enoxaparin sodium derived from porcine intestinal mucosa (Sanofi-Aventis U.S., Bridgewater, NJ) was applied to a Bio-Gel P-10 column (2.5X120 cm, Bio-Rad, Hercules CA) and eluted with 0.2 M $NH_4HCO_3$ at a flow rate of 14 ml/h. Elution of oligosaccharides was monitored by absorbance at 232 nm. $NH_4HCO_3$ was removed by heating in oven at 50˚C for 24 h.

### Preparation of the 6-*O*-desulfated Arixtra with MTSTFA

A detailed procedure on the preparation of 6-*O*-desulfated Arixtra was published previously [33]. Briefly, 4 mg of Arixtra was added to 10 volumes (w/w) of N-Methy-N-(trimethylsilyl)-trifluoroacetamide (MTSTFA, Sigma, ≥98.5%) and 100 volumes (v/w) of pyridine. The mixture was heated at 100˚C for 30 min, then quickly cooled in an ice-bath, followed by extensive dialysis and freeze-drying. The sample was resuspended in 50% acetonitrile/water at a concentration of 30 μM for later LC-MS/MS analysis.

## LC-MS/MS analysis

The 6-*O*-desulfated Arixtra (30 μM) was analyzed on a Thermo Orbitrap Fusion Tribrid (Thermo Fisher Scientific) coupled with an Ultimate 3000 Nano LC system (Dionex) using direct infusion. The flow rate was set to 1 μl/min. Mobile-phase was 50% acetonitrile. Nanoelectrospray voltage was set to 2.0 kV in negative ion mode. Full MS scan range was set to 200–2000 m/z at a resolution of 60,000, RF lens was 6%, and the automatic gain control (AGC) target was set to $2.0 \times 10^5$. For the MS/MS scans, the resolution was set to 50,000, the precursor isolation width was 3 m/z units, and ions were fragmented by collision-induced dissociation (CID) at a normalized collision energy of 80%.

## Cells

Mouse embryonic fibroblasts (MEF) and human foreskin fibroblasts (HFF) were cultured in Dulbecco's modified Eagle's medium (DMEM, Cellgro, Manassas, VA) containing 4.5 g/ml glucose, 10% fetal bovine serum (SAFC, Lenexa, KS), 1 mM sodium pyruvate, 2 mM L-glutamine, and 100 U/ml penicillin-streptomycin (Cellgro, Manassas, VA) at 37˚C with 5% $CO_2$. Mouse lung endothelial cells (WT, Hs3st1$^{-/-}$, Hs3st4$^{-/-}$, Hs3st1/4-double-knockout, Hs6st1$^{-/-}$, Hs6st2$^{-/-}$, and Hs6st1/2 double-knockout) were generated and reported in our recent study [34].

## Virus

MCMV (strain K181) was grown in MEF cells, while HCMV (Towne strain derived from Towne-BAC) was grown on HFF cells. Virus stock was prepared in 3X autoclaved milk, sonicated 3 times and stored at −80˚C. 3X autoclaved milk was prepared from Carnation (Nestle) instant nonfat dry milk powder. 10% milk was prepared in nano pure water, pH was adjusted to 7.0 and was autoclaved for 3 times. During infection, media was removed from the wells of cell culture plates and appropriately diluted virus stock was absorbed onto the cells in DMEM without serum. Cells were incubated for 1 hour with gentle shaking every 10 mins followed by washing 3X with PBS. Fresh complete medium was added and cells were incubated until the end point. For extracellular virus (ECV) purification, HFF were seeded in roller bottles, grown to confluency and infected with HCMV (Towne strain derived from Towne-BAC) at MOI of 0.01. Two days after 100% cytopathic effect was observed, infected cell medium was collected and centrifuged at low speed to pellet cellular debris, and the supernatant was transferred to new tubes and centrifuged at 20,000 g for 1 hour to pellet the ECV. This ECV pellet was re-suspended in phosphate buffer, sonicated to eliminate any aggregates, loaded over 15–50% continuous sucrose gradients and centrifuged in a SW-41 rotor at 39,000 RPM for 20 min. ECV bands were visualized in incandescent light and harvested by puncturing the sides of the centrifuge tubes. These bands were washed once with phosphate buffer, spun again and the final pellet resuspended in low salt phosphate buffer. An aliquot of the sample was used for assessment of initial quality of ECV by negative staining and transmission electron microscopy. Purified ECV were shipped on ice to Z biotech (Aurora, CO) for glycoarray binding analysis.

## Virus infection and cell viability assay

HFF cells plated in 12 well tissue culture plates were grown to confluency and pretreated for 1h with 10 μM concentration of candidate HS and then infected with HCMV (Towne strain derived from Towne-BAC) at a multiplicity of infection (MOI) of 3.0 or mock- infected in the presence of candidate HS. Five hundred μl of fresh complete medium containing HS was added to the wells on day 3 and day 6. For virus GFP enumeration assay, the cells were harvested at 5 days post infection and total number of GFP foci in three replicate wells for each

were counted. An average of >200 foci were present in mock-treated infected cells. At the designated time points, media was removed and cells were harvested by trypsinization. Cell viability was determined using trypan blue exclusion on TC20 automated cell counter (BioRad Laboratories, Hercules, CA) following manufacturer's protocol.

## Virus titers

Infected or mock-infected samples were harvested within the medium at the designated end points and stored at −80˚C before titration. In some experiments, media and cells were separated by low-speed (< 1000 × g) centrifugation and viral loads in supernatant and cells were quantified by titering on wild-type cells. Titers were performed as described earlier [35] with some modifications. In brief, monolayers of fibroblasts grown in 12 well plates and serial dilutions of sonicated samples were absorbed onto them for 1 h, followed by 3X washing with PBS. Carboxymethylcellulose (CMC) (Catalog No. 217274, EMD Millipore Corp., Billerica, MA) overlay with complete DMEM media (1-part autoclaved CMC and 3 parts media) was added and cells were incubated for 5 to 10 days. At the end point, overlay was removed and cells were washed 2X with PBS. Infected monolayers were fixed in 100% methanol for 7 min, washed once with PBS and stained with 1% crystal violet (Catalog No. C581-25, Fisher Chemicals, Fair Lawn, NJ) for 15 min. Plates were finally washed with tap water, air dried and plaques with clear zone were quantified.

## Immunoblots

Mouse lung endothelial cells (WT, Hs3st1$^{-/-}$, Hs3st4$^{-/-}$, Hs3st1/4-double-knockout, Hs6st1$^{-/-}$, Hs6st2$^{-/-}$, and Hs6st1/2 double-knockout) were infected with MCMV (K181 strain) and an MOI of 3.0 and the whole cell lysates were harvested at 2 hours post infection for analysis. The blot was probed with anti-IE1 antibody (catalog no. HR-MCMV-12, Center for Proteomics, University of Rijeka, Croatia) and HRP-conjugated goat anti mouse antibody (PI131444, Invitrogen) was used as the secondary antibody.

## Glycoarrays

A dilution series of purified HCMV virions were incubated on two different custom glycoarrays (Tables 1 and 2, Z-Biotech) using established protocols [36] and the arrays were analyzed to assess specific virus binding. Briefly, $10^5$ to $10^8$ pfu/ml of purified virions were incubated for an hour on glycoarrays containing six replicates of each glycosaminoglycan. After incubation, staining with primary antibody (mouse anti gB (clone 2F12, Virusys Inc, Taneytown, MD) was done at 100 μg/ml and secondary antibody (Goat anti mouse IgG AlexaFlour555) was done at 1μg/ml. Maximum strength fluorescent signal was obtained for $10^8$ pfu/ml concentration of the virus, therefore, only this concentration is represented in the final data obtained for plotting the graphs.

## Protein expression and purification

Merlin strain HCMV gB lacking the transmembrane region [37] and codon-optimized for expression in human cells, with an optimal Kozak sequence immediately 5' to the gene, was synthesized by Twist Biosciences (San Francisco, CA) and subcloned into pTwist CMV Beta-Globin WPRE Neoeukaryotic expression vector using the NotI and ClaI restriction sites. Purified plasmid was transfected into Expi293 cells (ThermoFisher). Culture supernatant was harvested seven days later and passed over a His Trap HP column (Cytiva) and gB protein was eluted with 450 mM imidazole. Imidazole was removed via buffer exchange using a centrifugal

**Table 1. A custom designed glycoarray containing hyaluronic acid, heparin, chondroitin sulfate and dermatan sulfate species.** The structure, molecular weight, number of sugar residues and sulfate groups per disaccharide for each glycosaminoglycan are listed.

| ID | Name | Structure and Molecular Weight | MW* | No. of Sugar Residues | Sulfate Groups per Disaccharide |
|---|---|---|---|---|---|
| GAG1 | Hyaluronic Acid dp10 (HA10) | ΔHexAβ1,3 [GlcNAcβ1,4 GlcAβ1,3]$_4$ GlcNAc | 1,950 | 10 | 0 |
| GAG2 | Hyaluronic Acid dp12 (HA12) | ΔHexAβ1,3 [GlcNAcβ1,4 GlcAβ1,3]$_5$ GlcNAc | 2,350 | 12 | 0 |
| GAG3 | Hyaluronic Acid dp14 (HA14) | ΔHexAβ1,3 [GlcNAcβ1,4 GlcAβ1,3]$_6$ GlcNAc | 2,700 | 14 | 0 |
| GAG4 | Hyaluronic Acid dp16 (HA16) | ΔHexAβ1,3 [GlcNAcβ1,4 GlcAβ1,3]$_7$ GlcNAc | 3,150 | 16 | 0 |
| GAG5 | Hyaluronic Acid dp18 (HA18) | ΔHexAβ1,3 [GlcNAcβ1,4 GlcAβ1,3]$_8$ GlcNAc | 3,650 | 18 | 0 |
| GAG6 | Hyaluronic Acid dp20 (HA20) | ΔHexAβ1,3 [GlcNAcβ1,4 GlcAβ1,3]$_9$ GlcNAc | 3,900 | 20 | 0 |
| GAG7 | Hyaluronic Acid Polymer (HA93) | ΔHexAβ1,3 [GlcNAcβ1,4 GlcAβ1,3]$_n$ GlcNAc | 93,000 | 462 | 0 |
| GAG8 | Heparin dp10 (H10) | ΔHexA,2S—GlcNS,6S-(IdoA,2S-GlcNS,6S)$_4$ | 3,000 | 10 | 3 |
| GAG9 | Heparin dp12 (H12) | ΔHexA,2S—GlcNS,6S-(IdoA,2S-GlcNS,6S)$_5$ | 3,550 | 12 | 3 |
| GAG10 | Heparin dp14 (H14) | ΔHexA,2S—GlcNS,6S-(IdoA,2S-GlcNS,6S)$_6$ | 4,100 | 14 | 3 |
| GAG11 | Heparin dp16 (H16) | ΔHexA,2S—GlcNS,6S-(IdoA,2S-GlcNS,6S)$_7$ | 4,650 | 16 | 3 |
| GAG12 | Heparin dp18 (H18) | ΔHexA,2S—GlcNS,6S-(IdoA,2S-GlcNS,6S)$_8$ | 5,200 | 18 | 3 |
| GAG13 | Heparin dp20 (H20) | ΔHexA,2S—GlcNS,6S-(IdoA,2S-GlcNS,6S)$_9$ | 5,750 | 20 | 3 |
| GAG14 | Heparin dp22 (H22) | ΔHexA,2S—GlcNS,6S-(IdoA,2S-GlcNS,6S)$_{10}$ | 6,300 | 22 | 3 |
| GAG15 | Heparin dp24 (H24) | ΔHexA,2S—GlcNS,6S-(IdoA,2S-GlcNS,6S)$_{11}$ | 6,850 | 24 | 3 |
| GAG16 | Heparin dp30 (H30) | ΔHexA,2S—GlcNS,6S-(IdoA,2S-GlcNS,6S)$_{14}$ | 9,000 | 30 | 3 |
| GAG17 | Chondroitin Sulfate AC dp10 (CS10) | ΔUA—(GalNAc,6S or 4S—GlcA)$_4$—GalNAc,6S or 4S | 2,480 | 10 | 1 |
| GAG18 | Chondroitin Sulfate AC dp12 (CS12) | ΔUA—(GalNAc,6S or 4S—GlcA)$_5$—GalNAc,6S or 4S | 2,976 | 12 | 1 |
| GAG19 | Chondroitin Sulfate AC dp14 (CS14) | ΔUA—(GalNAc,6S or 4S—GlcA)$_6$—GalNAc,6S or 4S | 3,472 | 14 | 1 |
| GAG20 | Chondroitin Sulfate AC dp16 (CS16) | ΔUA—(GalNAc,6S or 4S—GlcA)$_7$—GalNAc,6S or 4S | 3,968 | 16 | 1 |
| GAG21 | Chondroitin Sulfate AC dp18 (CSD18) | ΔUA—(GalNAc,6S or 4S—GlcA)$_8$—GalNAc,6S or 4S | 4,464 | 18 | 1 |
| GAG22 | Chondroitin Sulfate AC dp20 (CSD20) | ΔUA—(GalNAc,6S or 4S—GlcA)$_9$—GalNAc,6S or 4S | 4,960 | 20 | 1 |
| GAG23 | Chondroitin Sulfate D dp10 (CSD10) | ΔUA—(GalNAc,6S or 4S—GlcA +/- 2S)$_4$—GalNAc,6S | 2,480 | 10 | 1 or 2 |
| GAG24 | Chondroitin Sulfate D dp12 (CSD12) | ΔUA—(GalNAc,6S or 4S—GlcA +/- 2S)$_5$—GalNAc,6S | 2,976 | 12 | 1 or 2 |
| GAG25 | Chondroitin Sulfate D dp14 (CSD14) | ΔUA—(GalNAc,6S or 4S—GlcA +/- 2S)$_6$—GalNAc,6S | 3,472 | 14 | 1 or 2 |
| GAG26 | Chondroitin Sulfate D dp16 (CSD16) | ΔUA—(GalNAc,6S or 4S—GlcA +/- 2S)$_7$—GalNAc,6S | 3,968 | 16 | 1 or 2 |
| GAG27 | Chondroitin Sulfate D dp18 (CSD18) | ΔUA—(GalNAc,6S or 4S—GlcA +/- 2S)$_8$—GalNAc,6S | 4,464 | 18 | 1 or 2 |
| GAG28 | Chondroitin Sulfate D dp20 (CSD20) | ΔUA—(GalNAc,6S or 4S—GlcA +/- 2S)$_9$—GalNAc,6S | 4,960 | 20 | 1 or 2 |
| GAG29 | Dermatan Sulfate dp10 (DS10) | ΔHexA—GalNAc,4S—(IdoA—GalNAc,4S)$_4$ | 2,480 | 10 | 1 |
| GAG30 | Dermatan Sulfate dp12 (DS12) | ΔHexA—GalNAc,4S—(IdoA—GalNAc,4S)$_5$ | 2,976 | 12 | 1 |
| GAG31 | Dermatan Sulfate dp14 (DS14) | ΔHexA—GalNAc,4S—(IdoA—GalNAc,4S)$_6$ | 3,472 | 14 | 1 |
| GAG32 | Dermatan Sulfate dp16 (DS16) | ΔHexA—GalNAc,4S—(IdoA—GalNAc,4S)$_7$ | 3,968 | 16 | 1 |
| GAG33 | Dermatan Sulfate dp18 (DS18) | ΔHexA—GalNAc,4S—(IdoA—GalNAc,4S)$_8$ | 4,464 | 18 | 1 |
| GAG34 | Dermatan Sulfate dp20 (DS20) | ΔHexA—GalNAc,4S—(IdoA—GalNAc,4S)$_9$ | 4,960 | 20 | 1 |

*MW: Molecular weight (Dalton)

**Table 2. A custom designed glycoarray containing different heparan sulfate species.** The structure, molecular weight, number of sugar residues and sulfate groups per disaccharide for each glycosaminoglycan are listed.

| ID | Structure | MW* | No. of Sugar Residues | Sulfate Groups per Disaccharides |
|---|---|---|---|---|
| HS001 | GlcNAcα1-4GlcAβ1-4GlcNAcα1-4-GlcA | 1000 | 4 | 0 |
| HS002 | GlcAβ1-4GlcNAcα1-4GlcAβ1-4GlcNAcα1-4GlcA | 1,176 | 5 | 0 |
| HS003 | GlcNAcα1-4GlcAβ1-4GlcNAcα1-4GlcAβ1-4GlcNAcα1-4GlcA | 1,379 | 6 | 0 |
| HS004 | GlcAβ1-4GlcNAcα1-4GlcAβ1-4GlcNAcα1-4GlcAβ1-4GlcNAcα1-4GlcA | 1,555 | 7 | 0 |
| HS005 | GlcNAcα1-4GlcAβ1-4GlcNAcα1-4GlcAβ1-4GlcNAcα1-4GlcAβ1-4GlcNAcα1-4GlcA | 1,758 | 8 | 0 |
| HS006 | GlcAβ1-4GlcNAcα1-4GlcAβ1-4GlcNAcα1-4GlcAβ1-4GlcNAcα1-4GlcAβ1-4GlcNAcα1-4GlcA | 1,934 | 9 | 0 |
| HS007 | GlcNSα1-4GlcAβ1-4GlcNSα1-4GlcA | 1,076 | 4 | 1 |
| HS008 | GlcAβ1-4GlcNSα1-4GlcAβ1-4GlcNSα1-4GlcA | 1,252 | 5 | 0.8 |
| HS009 | GlcNSα1-4GlcAβ1-4GlcNSα1-4GlcAβ1-4GlcNSα1-4GlcA | 1,493 | 6 | 1 |
| HS010 | GlcAβ1-4GlcNSα1-4GlcAβ1-4GlcNSα1-4GlcAβ1-4GlcNSα1-4GlcA | 1,669 | 7 | 0.9 |
| HS011 | GlcNSα1-4GlcAβ1-4GlcNSα1-4GlcAβ1-4GlcNSα1-4GlcAβ1-4GlcNSα1-4GlcA | 1,910 | 8 | 1 |
| HS012 | GlcAβ1-4GlcNSα1-4GlcAβ1-4GlcNSα1-4GlcAβ1-4GlcNSα1-4GlcAβ1-4GlcNSα1-4GlcA | 2,087 | 9 | 0.9 |
| HS013 | GlcAβ1-4GlcNSα1-4GlcAβ1-4GlcNSα1-4GlcAβ1-4GlcNSα1-4GlcAβ1-4GlcNS6Sα1-4GlcA | 2,166 | 9 | 1.1 |
| HS014 | GlcAβ1-4GlcNSα1-4GlcAβ1-4GlcNSα1-4GlcAβ1-4GlcNS6Sα1-4GlcAβ1-4GlcNS6Sα1-4GlcA | 2,246 | 9 | 1.3 |
| HS015 | GlcAβ1-4GlcNSα1-4GlcAβ1-4GlcNS6Sα1-4GlcAβ1-4GlcNS6Sα1-4GlcAβ1-4GlcNS6Sα1-4GlcA | 2,327 | 9 | 1.6 |
| HS016 | GlcAβ1-4GlcNS6Sα1-4GlcAβ1-4GlcNS6Sα1-4GlcAβ1-4GlcNS6Sα1-4GlcAβ1-4GlcNS6Sα1-4GlcA | 2,406 | 9 | 1.8 |
| HS017 | GlcNSα1-4GlcAβ1-4GlcNSα1-4GlcAβ1-4GlcNSα1-4IdoA2Sβ1-4GlcNSα1-4GlcA | 1,990 | 8 | 1.3 |
| HS018 | GlcNSα1-4GlcAβ1-4GlcNSα1-4IdoA2Sβ1-4GlcNSα1-4IdoA2Sβ1-4GlcNSα1-4GlcA | 2,070 | 8 | 1.5 |
| HS019 | GlcNAcα1-4GlcAβ1-4GlcNSα1-4IdoA2Sβ1-4GlcNSα1-4IdoA2Sβ1-4GlcNSα1-4GlcA | 2,432 | 8 | 1.3 |
| HS020 | GlcNS6Sα1-4GlcAβ1-4GlcNS6Sα1-4GlcAβ1-4GlcNS6Sα1-4IdoA2Sβ1-4GlcNS6Sα1-4GlcA | 2,310 | 8 | 2.3 |
| HS021 | GlcNS6Sα1-4GlcAβ1-4GlcNS6Sα1-4IdoA2Sβ1-4GlcNS6Sα1-4IdoA2Sβ1-4GlcNS6Sα1-4GlcA | 2,389 | 8 | 2.5 |
| HS022 | GlcNAc6Sα1-4GlcAβ1-4GlcNS6Sα1-4IdoA2Sβ1-4GlcNS6Sα1-4IdoA2Sβ1-4GlcNS6Sα1-4GlcA | 2,353 | 8 | 2.3 |
| HS023 | GlcNS6Sα1-4GlcAβ1-4GlcNS3S6Sα1-4IdoA2Sβ1-4GlcNS6Sα1-4GlcA | 1,893 | 6 | 2.7 |
| HS024 | GlcNAc6Sα1-4GlcAβ1-4GlcNS3S6Sα1-4IdoA2Sβ1-4GlcNS6Sα1-4IdoA2Sβ1-4GlcNS6Sα1-4GlcA | 2,433 | 8 | 2.5 |

*MW: Molecular weight (Dalton)

filter device with a 50,000 dalton cutoff (Pall Corp.). Protein purification was verified by SDS-PAGE and protein concentration was measured by BCA (ThermoScientific) (S5 Fig).

## Preparation of heparin coated Surface Plasmon Resonance (SPR) sensor chip

gB protein purification and characterization of heparin analogs have been described above. Porcine intestinal heparin (15 kDa) was purchased from Celsus Laboratories (Cincinnati, OH) and *N*-desulfated heparin (15 kDa) and 6-*O*-desulfated heparin (15 kDa) were from Galen Lab Supplies (North Haven, CT). Sensor SA chips were from Cytiva Life Sciences (Uppsala, Sweden). SPR measurements were performed on a Biacore T200 operated using Biacore T200 control and T200 Evaluation software (version 3.2). Biotinylated heparin was prepared by conjugating its reducing end to amine-PEG3-Biotin (Pierce, Rockford, IL). In brief, heparin (2

mg) and amine-PEG3-Biotin (2 mg, Pierce, Rockford, IL) were dissolved in 200 μl $H_2O$, 10 mg $NaCNBH_3$ was added. The reaction mixture was heated at 70˚C for 24 h, after that a further 10 mg $NaCNBH_3$ was added and the reaction was heated at 70˚C for another 24 h. After cooling to room temperature, the mixture was desalted with the spin column (3,000 MWCO). Biotinylated heparin was collected, freeze-dried and used for SA chip preparation. The biotinylated heparin was immobilized to streptavidin (SA) chip based on the manufacturer's protocol. The successful immobilization of heparin was confirmed by the observation of ~200 resonance unit (RU) increase on the sensor chip. The control flow cell (FC1) was prepared by 1 min injection with saturated biotin.

**Kinetic measurement of interaction between heparin and gB protein using Biacore.** The gB protein was diluted in HBS-EP+ buffer (0.01 M HEPES, 0.15 M NaCl, 3 mM EDTA, 0.05% surfactant P20, pH 7.4). Different dilutions of protein samples were injected at a flow rate of 30 μL/min. At the end of the sample injection, the same buffer was flowed over the sensor surface to facilitate dissociation. After a 3 min dissociation time, the sensor surface was regenerated by injecting with 30 μL of 2M NaCl to get fully regenerated surface. The response was monitored as a function of time (sensorgram) at 25˚C.

**Solution competition study between heparin on chip surface and heparin analogs in solution using SPR.** gB protein (250 nM) mixed with heparin (1000 nM) or heparin analogs (1000 nM) in HBS-EP+ buffer were injected over heparin chip at a flow rate of 30 μL/min, respectively. After each run, the dissociation and the regeneration were performed as described above. For each set of competition experiments on SPR, a control experiment (only protein without any heparin) was performed to make sure the surface was completely regenerated and that the results obtained between runs were comparable.

## Statistics

Data was analyzed by ordinary one way ANOVA with multiple comparisons comparing the means of each test sample with control, and corrected using Dunnett's post hoc test. In some cases, pairwise comparisons were made using Student's t-test. Differences were considered significant if $p < 0.05$. All statistical tests were done in Prism (Prism version 8.0, GraphPad Software, San Diego, California USA, www.graphpad.com) Standard error of mean or standard deviation was plotted as error bars. An asterisk ($^*$) indicates significant differences compared to mock or wild-type.

## Results

### Purified HCMV extracellular virions preferentially bind to sulfated glycosaminoglycans with increased degree of polymerization

First, we sought to establish the category of GAG that preferentially binds to purified HCMV virions. HCMV extracellular virions were purified as described above and incubated with custom glycoarrays containing increasing molecular weight species of hyaluronic acid, heparin, chondroitin sulfate, and dermatan sulfate (Table 1). HCMV binding to non-sulfated hyaluronic acid (GAG1-GAG7) was negligent but significant binding to all heparin species was detected with a trend of increased binding to heparins as their *dp* increased (Fig 1A). HCMV also bound to large size chondroitin sulfate D (GAG28, *dp20*), and dermatan sulfate oligosaccharides (GAG32-GAG34, *dp16-dp20*) but not to chondroitin sulfate AC (GAG17-GAG22). It is important to note that while the chondroitin sulfate A (CS-A) is sulfated at C4 of the Gal-NAc, and the chondroitin sulfate C (CS-C) is sulfated at the C6 of the GalNAc only, the chondroitin sulfate D is sulfated at C2 of the glucuronic acid as well as the C6 of the GalNAc sugar

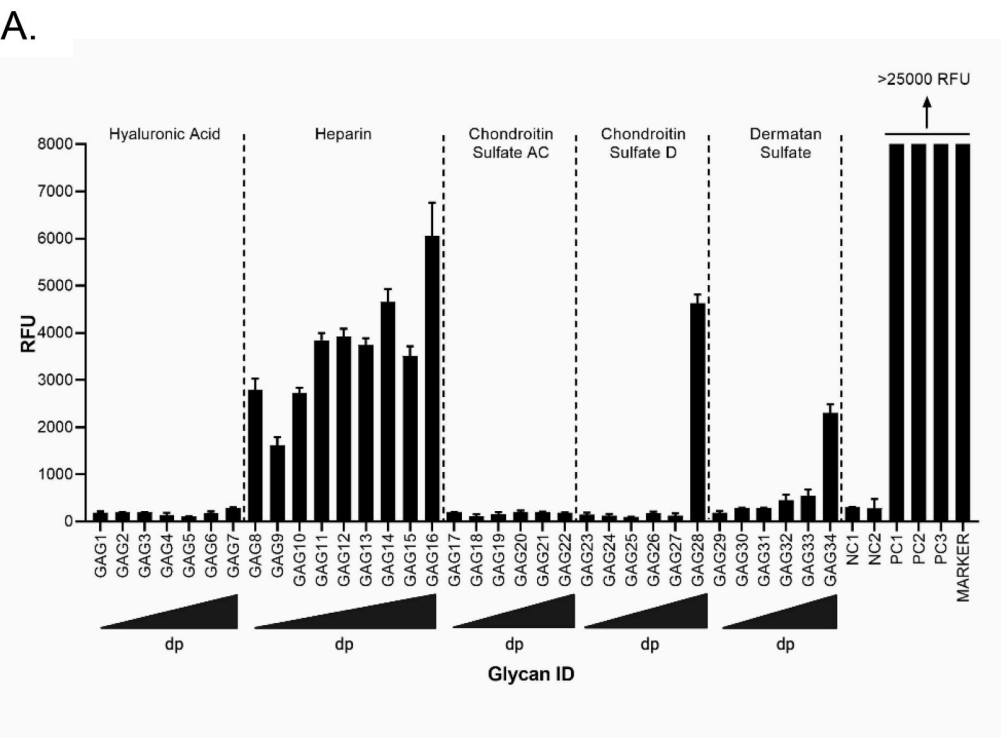

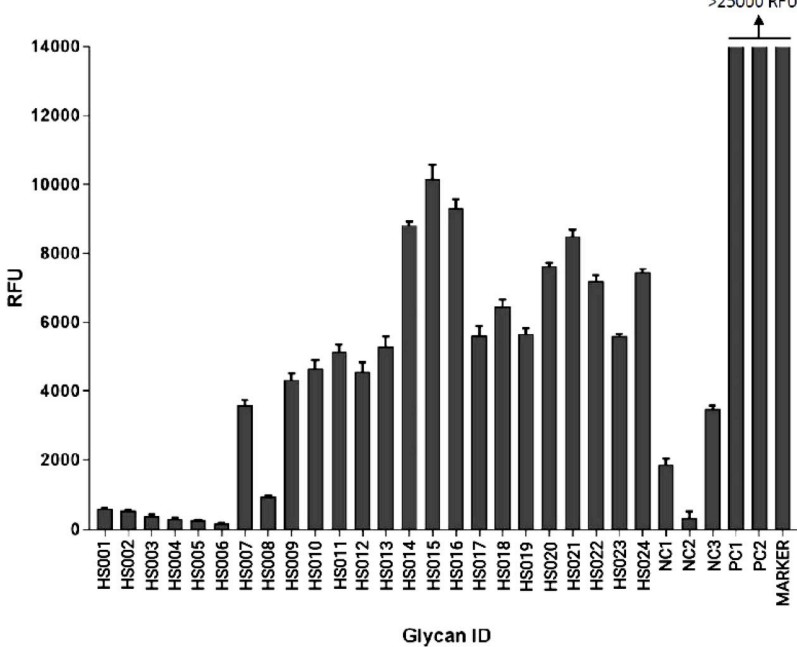

**Fig 1. Binding of human cytomegalovirus on glycosaminoglycan glycoarrays. (A)** Binding of purified extracellular CMV virions on a custom designed glycosaminoglycan glycoarray. Relative fluorescence units (RFU), which are directly proportional to the amount of virus binding, are plotted on the Y-axis in the graph. Ligand descriptions and chain structures are provided in Table 1. Six replicates for each GAG were used in the assay. NC1: Negative control 1 (print buffer), NC2: (Biotinylated Glycan), PC1: positive control; human IgG (0.1 mg/ml), PC2: mouse IgG (0.1 mg/ml), PC3: rabbit IgG (0.1 mg/ml), *dp*: degree of polymerization, triangles at the bottom of the graph represent an

increasing degree of polymerization of GAGs from the left to the right. **(B)** Binding of purified extracellular CMV on a custom designed heparan sulfate glycoarray. Relative fluorescence units (RFU) are plotted on the Y-axis in the graph. Ligand descriptions and chain structures are provided in Table 2. Six replicates for each ligand were used. NC1: negative control 1 (print buffer), NC2: (biotinylated glycan), NC3: human IgG (0.1 mg/ml), PC1: mouse IgG (0.1 mg/ml), PC2: rabbit IgG (0.1 mg/ml). Data was analyzed by ordinary one way ANOVA with multiple comparisons comparing the means of each test with control, and corrected using Dunnett's post hoc test, showing significant differences among means (p < 0.0001). Standard deviation was plotted as error bars.

and hence has double the amount of sulfation compared to CS-A and CS-C. Dermatan sulfate (DS), formerly referred to as CS-B, is formed from the polymer backbone of chondroitin sulfate by the action of chondroitin-glucuronate C5 epimerase, which epimerizes individual d-glucuronic acid residues to L-iduronic acid. The binding affinity to DS was also size-dependent increasing from GAG32-GAG34 (*dp16-dp20*). Heparin (*dp30*) was the best HCMV binder in this assay. The positive and negative controls worked as expected.

On a second HS specific array (Table 2), HCMV showed strong binding to the HS with longer monosaccharide chains (HS007 to HS024) and minimal binding to unsulfated glycans (HS001-HS006) (Fig 1B). The maximum binding was observed for HS014, HS015 and HS016, which are all 6-*O*-S 9-mers with moderate amount of sulfation (1.3–1.8 sulfate group per disaccharide). Also, significant amount of binding was observed for 2-*O*-S (HS17-HS19), 6-*O*-S/2-*O*-S (HS20-22) and 2-*O*-S/6-*O*-S/3-*O*-S (HS23-24) HS that had high amount of sulfation (1.3–2.7 sulfate group per disaccharide) and 6–8 disaccharides per chain. Overall the data from these experiments indicate that the *dp* of HS as well as sulfation is important for HCMV binding.

## The degree of polymerization of GAG chains impacts CMV infectivity

Glycosaminoglycans of different *dp* were fractionated from enoxaparin (a low molecular weight heparin). All of these GAGs are based on a HS backbone and differ in either *dp* or degree/place of sulfation or both (Fig 2 and S1 Fig). These GAGs, along with heparin and Arixtra (fondaparinux sodium), were first screened in a GFP-based preliminary virus focus reduction assay using GFP tagged HCMV (Towne strain). The viral GFP expression was most efficiently reduced by heparin salt (PIHSS; Heparin sodium salt from porcine intestinal mucosa) whereas Arixtra, 6-*O*-desulfated Arixtra and enoxaparin had little to no impact on GFP expression (Fig 2). In general, enoxaparin derived GAGs with higher *dp* were more efficient in reducing viral GFP compared to low *dp* derivatives. To follow up on this primary GFP based screening, we performed viral titer assay using HCMV (Towne strain) that measures total virus yields at 5 days post-infection. Most reduction in viral titers was observed for heparin (PIHSS) followed by enoxaparin derivative with >20 *dp* (Fig 3A). Plotting of viral titer reduction as a function of *dp* revealed a general trend where higher *dp* derivatives lead to higher reduction in viral titers (Fig 3B). Thus, this experiment indicated that longer HS chains are more efficient at reducing HCMV titers in cells. To investigate whether this inhibitory effect was due to an increase in the number of HCMV binding sites per chain of longer chain GAGs towards virus particles, the experiments were repeated at 0.05 g/L concentrations of GAGs instead of the previously used molar equivalent concentrations (Fig 3C). As micromolar concentration (10 μM) of GAGs is based on number of molecules provided, GAGs consisting of longer chain will have more potential binding sites for virus than those of shorter chains. The other concentration (0.05 g/L) is based on weight; thus this concentration normalizes the number of potential virus binding sites for GAGs consisting of both long and short chains. Interestingly, similar trend of inhibitory results leaning towards efficacy of higher *dp* against HCMV infection were obtained at 0.05 g/L indicating that this effect is not merely due to a

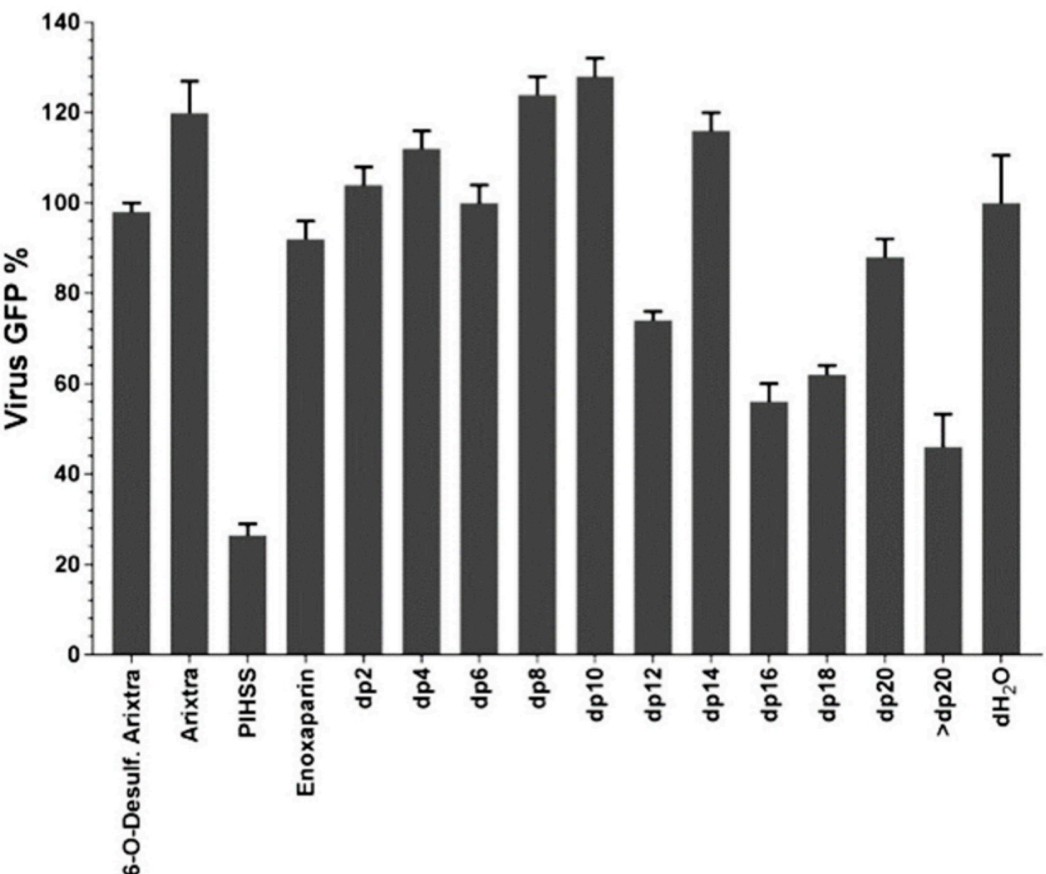

**Fig 2. Inhibition of HCMV growth by glycosaminoglycan derivatives.** Primary human foreskin fibroblasts (HFF) grown in 96 well plate were pretreated for one hour with 10 μM of 1) 6-*O*-desulfated Arixtra, 2) Unmodified Arixtra, 3) Heparin sodium salt from porcine intestinal mucosa (PIHSS), 4) Enoxaparin, or series of heparin oligosaccharide from enoxaparin: 5) *dp2*, 6) *dp4*, 7) *dp6*, 8) *dp8*, 9) *dp10*, 10) *dp12*, 11) *dp14*, 12) *dp16*, 13) *dp18*, 14) dp20 15) > *dp20* or control (dH$_2$O). Cells were infected with GFP tagged HCMV (Towne strain) virus at an MOI of 3.0 in the presence of the test glycosaminoglycans, which were maintained in the cell culture medium. At 5 days post-infection, cells were fixed and number of foci (GFP) was counted in triplicate well for each sample under an epifluorescent microscope. An average of >200 foci were present in mock-treated infected cells. Percent of viral GFP was calculated compared to virus only infected control (100% GFP expression). Results are representative of three independent replicates. Standard error of mean was plotted as error bars. Data was analyzed by ordinary one way ANOVA with multiple comparisons comparing the means of each test with control, and corrected using Dunnett's post hoc test, showing significant differences among means (p <0.0001).

higher number of potential independent binding sites in the longer GAG chains and instead involves a difference in the molecular interaction between HCMV and the longer GAG chains. A line graph for each concentration of GAGs was generated that demonstrates the relationship of viral titer and degree of polymerization (Fig 3D). Although GAG treatment is not known to induce cell death, to rule out that these effects on virus titers could be attributed to the health of cells, we performed cell viability assays in both uninfected and infected settings. Cell viability was not affected at the treated concentrations of any of our test GAGs (S4A Fig). Moreover, heparin (PIHSS) and some enoxaparin derivatives (*dp* 12 or greater) appeared to protect cells from death that was evident in mock-treated controls (S4B Fig). These results corroborate the results of our glycoarray experiments that showed that GAG with higher *dp* have higher CMV binding compared to GAG with lower *dp* (Fig 1A).

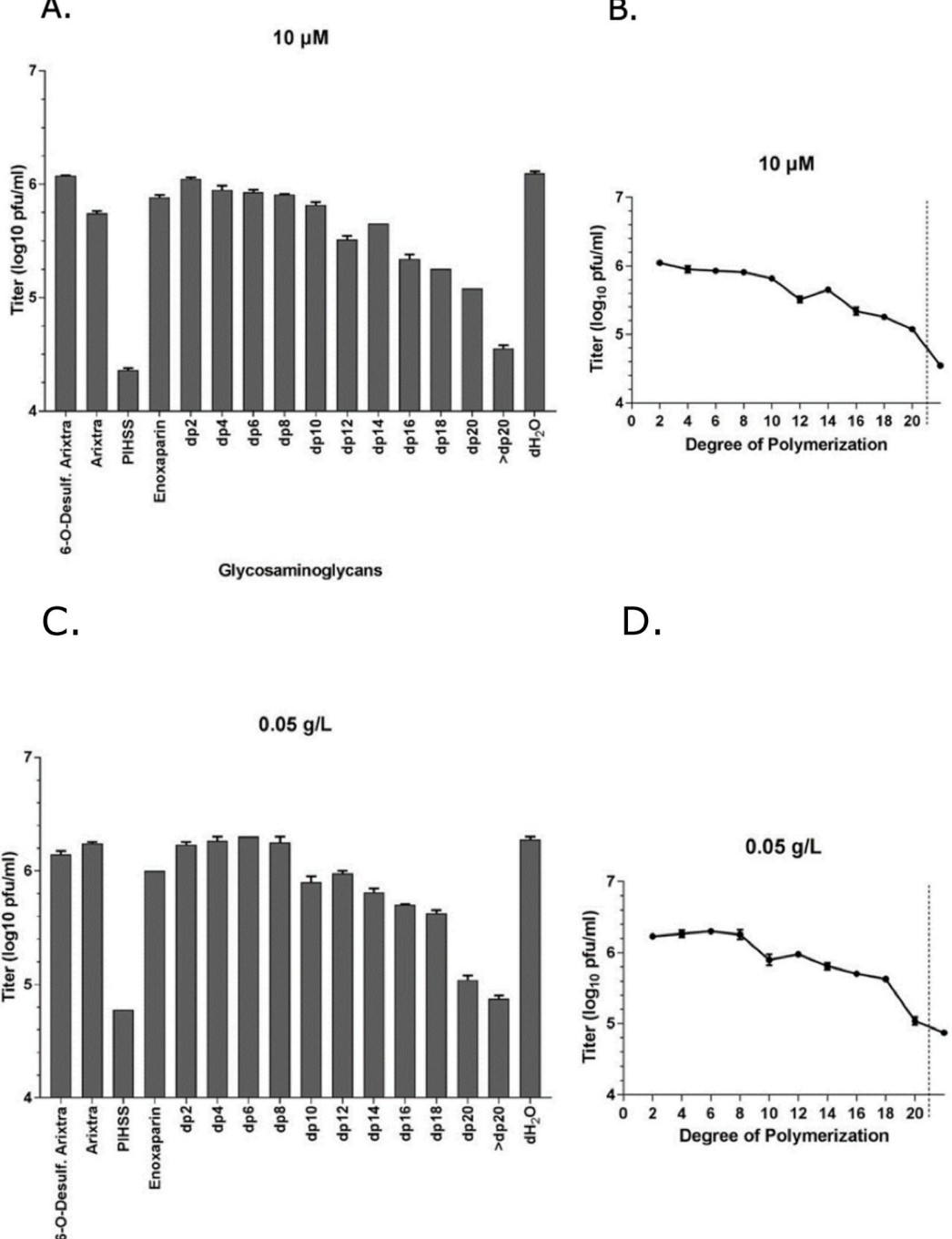

**Fig 3. Effect of glycosaminoglycan derivatives on HCMV growth. (A)** Primary human foreskin fibroblasts (HFF) were pretreated for one hour with 10 μM of 1) 6-*O*-desulfated Arixtra, 2) Regular Arixtra, 3) Heparin sodium salt from porcine intestinal mucosa (PIHSS), 4) Enoxaparin, or series of heparin oligosaccharide from enoxaparin: 5) *dp2*, 6) *dp4*, 7) *dp6*, 8) *dp8*, 9) *dp10*, 10) *dp12*, 11) *dp14*, 12) *dp16*, 13) *dp18*, 14) *dp20* 15) > *dp20* or control (dH₂O). Cells were infected with HCMV (Towne strain) virus at an MOI of 3.0 in the presence of test glycosaminoglycans. Cells and media were harvested at 5 days post-infection and titered for HCMV plaque forming units (pfu) on fresh fibroblasts in tissue culture dishes. Individual samples (3 replicates each) were quantified and displayed as total pfu/ml on Y-axis. **(B)** Virus titer is plotted (Y-axis) against degree of polymerization (X-axis). Data points ahead of the broken line is for a mixture of GAGs (*dp*>20). Results are representative of three independent replicates. Standard error of mean was plotted as error bars. **(C)** Primary human foreskin fibroblasts (HFF) were pretreated for one hour with 0.05 g/L of 1) 6-*O*-desulfated Arixtra, 2) Regular

Arixtra, 3) Heparin sodium salt from porcine intestinal mucosa (PIHSS), 4) Enoxaparin, or series of heparin oligosaccharide from enoxaparin: 5) *dp2*, 6) *dp4*, 7) *dp6*, 8) *dp8*, 9) *dp10*, 10) *dp12*, 11) *dp14*, 12) *dp16*, 13) *dp18*, 14) *dp20* 15) > *dp20* or control (dH$_2$O). Cells were infected with HCMV (Towne strain) virus at an MOI of 3.0 in the presence of test glycosaminoglycans. Cells and media were harvested at 5 days post-infection and titered for HCMV plaque forming units (pfu) on fresh fibroblasts in tissue culture dishes. Individual samples (3 replicates each) were quantified and displayed as total pfu/ml on Y-axis. **(D)** Virus titer is plotted (Y-axis) against degree of polymerization (X-axis). Data points ahead of the broken line is for a mixture of GAGs (*dp*>20). Results are representative of three independent replicates. Standard error of mean was plotted as error bars. Data was analyzed by ordinary one way ANOVA with multiple comparisons comparing the means of each test with control, and corrected using Dunnett's post hoc test, showing significant differences among means (p <0.0001).

## Cell lines defective in expression of specific sulfation enzymes have reduced CMV titers and reduced virus entry

Due to species specificity of HCMV, animal models are frequently used to study CMV pathogenesis [38,39]. Studies of murine CMV (MCMV) infections of mice have served a major role as a model of CMV biology and pathogenesis [40]. Lung endothelial cell lines from adult mice were mutated for specific sulfotransferase enzymes by a conditional Cre-LoxP or a CRISPR-Cas9 based gene editing system [34,41,42]. Since previous studies showed that 3-*O*-S HS is important for HCMV entry in human iris stromal cells [43], we analyzed virus replication in Hs3st1 and Hs3st4 (Glucosaminyl 3-*O*-sulfotransferase 1 and 4, respectively) knockout cell lines as well as the Hs3st1/4 double knockout cell line. At high (5.0) as well as low (0.01) multiplicity of infection (MOI), MCMV growth was significantly reduced in the single Hs3st1 and Hs3st4 knockouts as well as in the double Hs3st1/4 knockouts, indicating that 3-*O*-sulfation of HS is important for HCMV infection (Fig 4). Further, we probed whether virus entry is impacted in the cells knocked out for different combinations of sulfotransferases. Expression of viral immediate early protein (IE1) has been used as a surrogate for virus entry since it's one of the earliest events after a successful virus entry [44–46]. Results of an immunoblot probing for IE1 show that virus entry is significantly reduced in Hs3st1[-/-], Hs3st4[-/-], Hs3st1/4-double-knockout, Hs6st1[-/-], Hs6st2[-/-], and Hs6st1/2 double-knockout cells, compared to the wild-type cells (Fig 5A and 5B).

## Kinetics measurement of gB-heparin interactions using SPR

HCMV glycoprotein B (gB) was expressed in mammalian cells, purified and used in GAG binding assays utilizing SPR. Sensorgrams of gB-heparin interaction are shown in Fig 6A. Concentrations of gB used were 1000, 500, 250, 125, and 63 nM, respectively from top to bottom. The black curves are the fitting curves using T200 Evaluation software (version 3.2). The SPR results showed the kinetic of gB-heparin interaction: association rate constant: ka = 9.8× $10^3$ (±68) *(1/MS)*, dissociation rate constant: kd = 4.2×$10^{-4}$ (±1.7 ×$10^{-6}$) *(1/S)*; and binding equilibrium dissociation constant ($K_D$ = kd/ka): $K_D$ = 4.3×$10^{-8}$ (M).

**Solution competition study on the interaction between heparin (on surface) with gB added with heparin or heparin analogs in solution.** To examine the effect of heparin structure on heparin-gB interaction, solution/surface competition experiments were performed by SPR. Heparin (1000 nM) or heparin analogs (1000 nM) in HBS-EP+ buffer were pre-mixed with gB were injected. The results show (Fig 6B and 6C) the heparin in concentration of 1000 nM completely inhibited the gB binding to surface heparin. The inhibition was greatly reduced with desulfation of heparin.

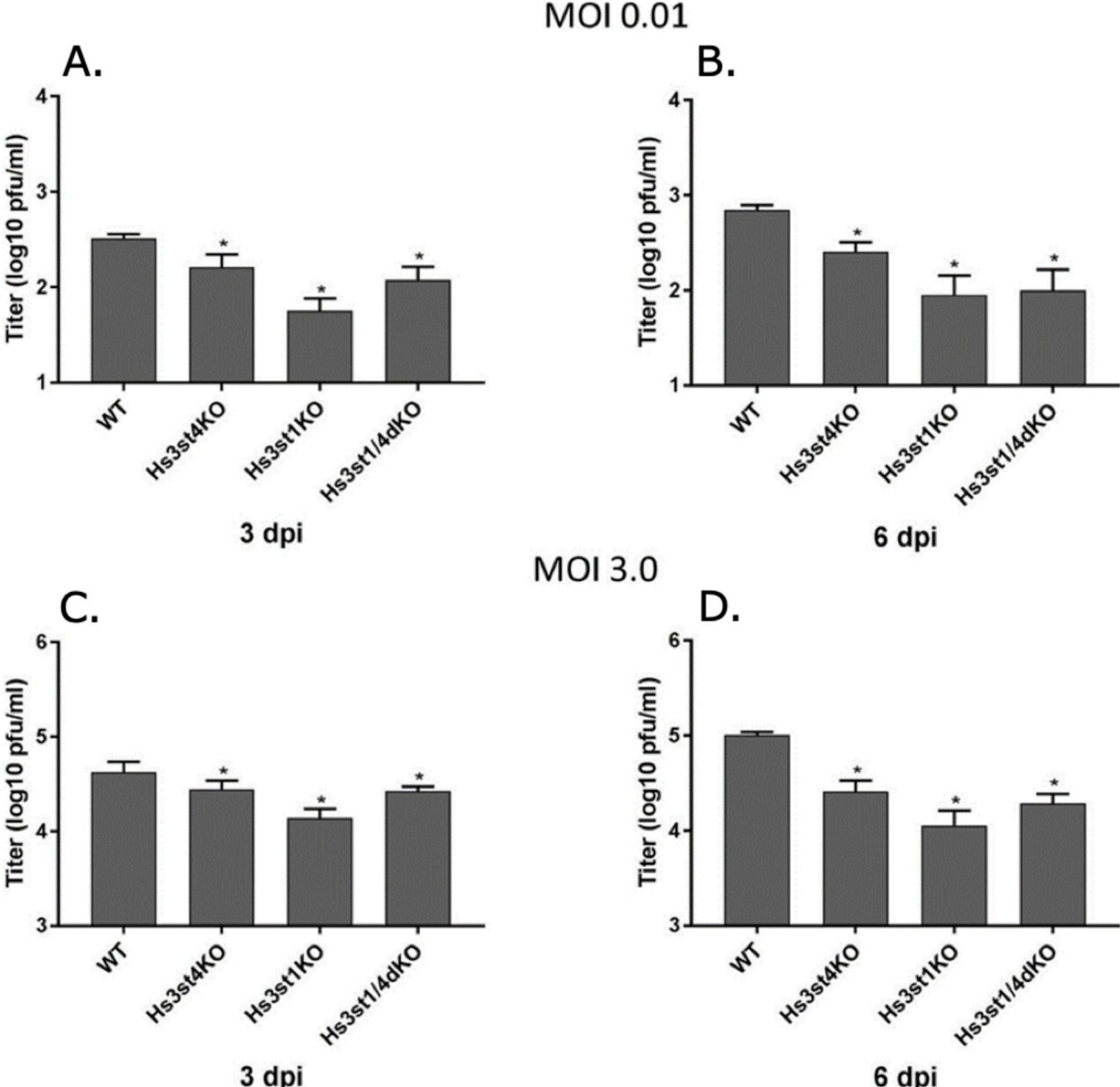

**Fig 4. Mouse CMV replication in sulfotransferase knockout cell lines.** Cells were grown to 90% confluency and infected with wild-type MCMV (strain K181) at low (0.01, **(A), (B)** and high (3.0) **(C), (D)** MOI. Cells and the medium were harvested at 3- and 5-days post-infection, sonicated to release the virus and diluted for plating on to wild-type MEF in tissue culture dishes in order to enumerate total MCMV pfu/ml. Results are representative of three independent replicates. Data was analyzed by ordinary one way ANOVA with multiple comparisons comparing the means of each test with control, and corrected using Dunnett's post hoc test, showing significant differences among means (p <0.05). Standard error of mean was plotted as error bars. An asterisk (*) indicates significant inhibition compared to wild-type. Hs3st1 and Hs3st4: Glucosaminyl 3-*O*-sulfotransferase 1 and 4, respectively. WT: wild type; KO: knockout.

## Discussion

In this study, we utilized multiple approaches, including glycoarray binding analysis, HS mimics, HS mutant cell lines, and gB-heparin binding to demonstrate that specifically sulfated HS with higher degree of polymerization affect CMV infection and binding. The results significantly advance the age-old knowledge of HS binding to herpesviruses by illustrating the

A.

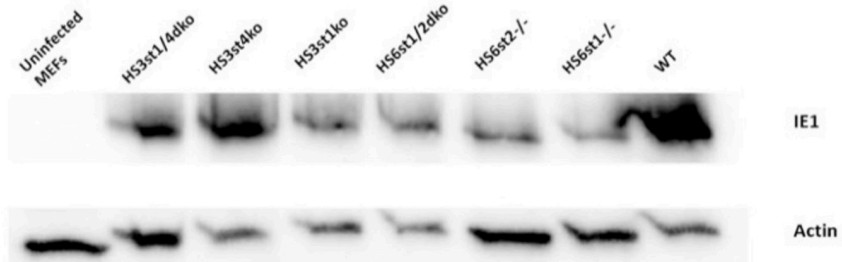

B.

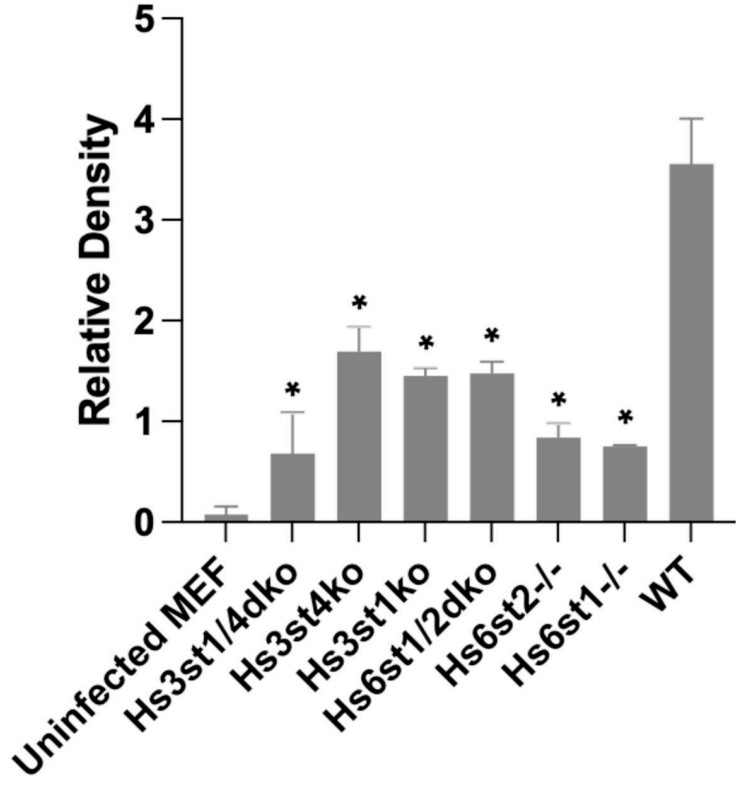

**Fig 5. Mouse CMV entry in sulfotransferase knockout cell lines. (A)** Cells were grown to 90% confluency and infected with wild-type MCMV (strain K181) at an MOI of 3.0. Cells were harvested at 2 hours post-infection, and whole cell lysates were loaded on a polyacrylamide gel for blotting. The blots were probed with anti IE1 antibody. Beta-actin was used as a loading control. **(B)** Bands from two independent experiments were quantified by densitometry and means were plotted. Standard error of mean was plotted as error bars. Hs6st1 and Hs6st4: Glucosaminyl 6-*O*-sulfotransferase 1 and 4, respectively. Hs3st1 and Hs3st4: Glucosaminyl 3-*O*-sulfotransferase 1 and 4, respectively. WT: wild type; dko: double knockout. Data was analyzed by ordinary one way ANOVA with multiple comparisons

comparing the means of each test sample with control, and corrected using Dunnett's post hoc test. Differences were considered significant (*) if p<0.05.

importance of HS structural modifications in CMV binding and infection. We first screened several sulfated or unsulfated GAGs with complex sugar structure to investigate which GAGs are more efficient at binding to HCMV virions. This glycoarray analysis indicated that HCMV bound heparins with strong affinity and showed increased affinity for longer chain length heparins (Fig 1A). Binding of CS was limited to the highest dp of only CS-D, a CS chain that is primarily sulfated at the C-2 position of GlcA and the C-4 position of GalNAc. CS-D has a much higher sulfate density compared to CS-A/CS-C (which are primarily only monosulfated at the GalNAc at the C-4 and C-6 positions, respectively), and the 2-O-sulfation of the GlcA is similar to the common 2-O-sulfation of IdoA found in heparin. Dermatan sulfate also only bound significantly in the highest dp tested. While DS has a sulfate density similar to CS-A/CS-C (one sulfo group per disaccharide, found at the GalNAc), it also bears similarity to heparin in that it has IdoA in its disaccharide repeat as most commonly found in heparin, unlike CS which has GlcA (differing by epimerization at the C-5 position). We would hypothesize that it is these heparin-like qualities that causes CS-D and DS to bind at high dp.

To further investigate this binding, we utilized another glycoarray consisting of HS of varied polymerization and sulfation levels. The results from this glycoarray indicated that HCMV binds strongly with HS having both longer monosaccharide chain and a moderate level of sulfation (Fig 1B). Thus, sulfated HS with more complex branches and sulfation patterns preferentially bind to HCMV. Next, we fractioned HS by length (dp 2–20) from enoxaparin and tested their ability to inhibit HCMV growth in cell culture by competing with HCMV binding. The GFP tagged HCMV was used and the number of GFP+ foci was quantified in the presence of increasing HS chain length. Amounts of viral GFP was more effectively reduced when cells were pretreated and maintained with GAGs having a higher dp (Fig 2). This assay served as a surrogate for a virus entry assay since the GFP is independently expressed from an early promoter in the virus genome [47]. For a deeper understanding of this reduction, we performed a similar experiment where HCMV Towne strain was used and viral load was quantified at 5 days post-infection (Fig 3). Significant reduction in virus titers was observed in samples treated with higher dp of GAG but not with lower dp corroborating the results from glycoarray experiments that chain length of GAG is an important factor in determining HCMV binding. Also, this effect was not due to a simple increase in the number of potential HCMV binding sites per mole of GAG, as evidenced by similar trend of inhibition obtained when treating cells with equivalent µM or g/L concentrations of GAGs. Treatment of cells with these GAGs did not affect cell viability for the duration of treatment (S4A Fig) confirming that the observed reduction in virus titer was not due to the cell death. Moreover, cells pretreated and maintained with GAGs of longer dp resisted infection induced cell death at late time post-infection (S4B Fig). We also tested the impact of specific HS sulfation mutants on MCMV infection. As 3-O-sulfation has been reported to be critical for herpesvirus entry [43,48], we tested MCMV growth in Hs3st1, Hs3st4 and dual Hs3st1/4 knockout cells. For both high and low MOI, virus titer was significantly reduced in Hs3st1, Hs3st4 and dual Hs3st1/4 knockout cells (Fig 4). Although the differences were statistically significant, we did not see a robust inhibition of virus titers in these assays compared to the wild-type. To directly assess the impact of HS sulfation on virus entry, we used several mutant cell lines deficient in HS sulfation enzymes. All of these cells were defective in virus entry as assessed by IE1 protein expression (Fig 5). The GAG experiments used a fibroblast cell culture system and a fibroblast tropic strain of HCMV (Towne),

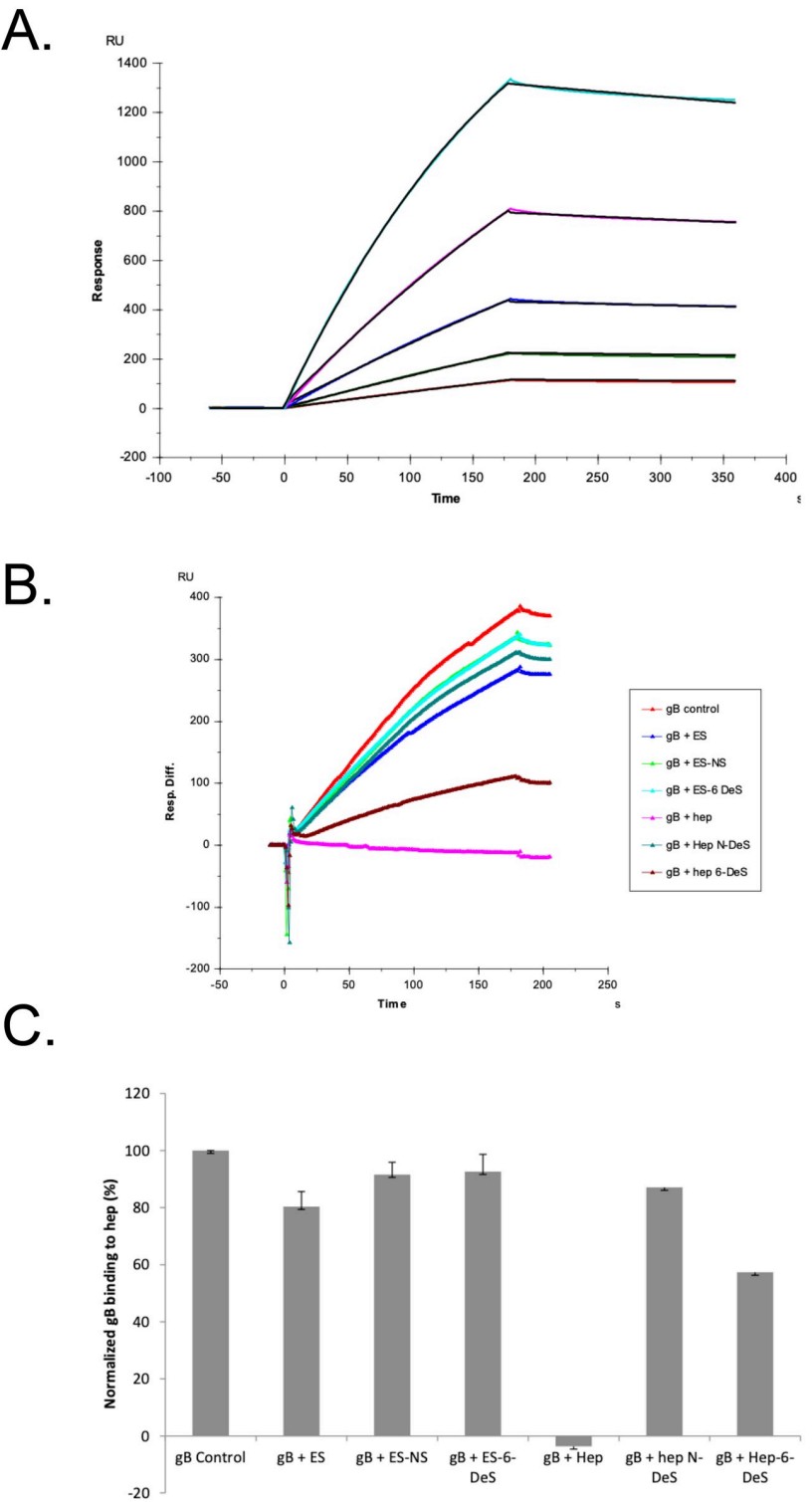

**Fig 6. gB-heparin interaction. (A)** SPR sensorgrams of gB-heparin interaction. Concentrations of gB (from top to bottom): 1000, 500, 250, 125, and 63 nM, respectively. The black curves are the fitting curves generated using the T200 Evaluation software (version 3.2). **(B)** Sensorgrams of solution heparin or analogs /surface heparin competition. gB concentration was 250 nM, and concentrations of heparin/analogs in solution were 1000 nM. **(C)** Bar graphs (based on triplicate experiments with standard deviation) of normalized gB binding preference to surface heparin by competing with heparin or heparin analogs in solution. ES: Enoxaparin Sodium; ES-NS: Enoxaparin Sodium *N*-sulfated; ES-6 DeS: Enoxaparin Sodium 6-*O*-desulfated; Hep: Heparin.

whereas MCMV experiments used lung epithelial cells, thus providing the experimental data from multiple cell types and two different viruses. While the GFP based assays provide a surrogate for virus entry assays, the real impact of virus entry inhibitors would be a reduction in viral titers at the end of infection since an entry inhibitor that only delays virus entry would be of little translational value. Thus virus yield and titers were used a measure of effectiveness of GAG inhibitors.

The enzymatic modification of HS chains is known to generate unique binding sites for viral ligands. For example, 3-*O*-sulfation modification in HS chain generates fusion receptor for HSV glycoprotein D (gD) promoting viral entry and spread [49]. The 3-*O*-S HS is a product of enzymatic modification at C3 position of glucosamine residue, which is relatively rare in comparison to other HS modifications. Expression of Hs3st can make normally resistant Chinese hamster ovary (CHO-K1) cells susceptible to HSV-1 infection [50]. Studies in clinically relevant primary human corneal fibroblasts have also shown 3-*O*-S HS as a primary attachment receptor for HSV entry [48]. Interestingly, both HSV-1 and HSV-2 use HS as an attachment receptor but HSV-1 binds to distinct modification sites on HS that HSV-2 is unable to, which could explain some of the differences in cell tropism exhibited by these two viruses [51]. For example, while N-sulfation and carboxyl groups are required for both HSV-1 and HSV-2 binding, only HSV-1 is able to bind the specific modification sites generated by 2-*O*-, 6-*O*-, and 3-*O*-sulfations [52]. The *O*-desulfated heparins have little or no inhibitory effect on HSV-1 infection but inhibit HSV-2 infection. This susceptibility to *O*-desulfated heparins can be transferred to HSV-1 by recombinant transfer of the gene for glycoprotein C (gC-2) from HSV-2 [52]. We reported earlier that 3-*O*-S HS are important for HCMV entry in human iris stromal (HIS) cells [43]. The expression of Hs3st in HIS cells promoted HCMV internalization, while pretreatment of HIS cells with heparinase enzyme or treatment with anti-3-*O*-S HS (G2) peptide significantly reduced HCMV plaques/foci formation. In addition, co-culture of the HCMV-infected HIS cells with CHO-K1 cells expressing 3-*O*-S HS significantly enhanced cell fusion. A similar trend of enhanced fusion was observed with cells expressing HCMV glycoproteins (gB, gO, and gH-gL) co-cultured with 3-*O*-S HS cells. These results highlight the role of 3-*O*-S HS during HCMV entry.

Herpesvirus glycoprotein B is one of the most conserved glycoprotein across the herpesvirus family [53]. Herpes Simplex Virus -1 (HSV-1) gB and gC have been shown to be responsible for HS binding on cell surface [8,54]. However, HCMV is not known to possess a sequence homolog of gC. Thus, we expressed and purified HCMV gB and used it in a GAG binding assay using SPR. This gB showed a very strong binding to heparin. Upon competition with different heparin analogs, it was found that only heparin itself and a 6-desulfated form of heparin significantly competed with gB-bound heparin. Enoxaparin and its desulfated homolog were unable to compete with heparin. These results confirm the results in our GAG screening assay where the sulfated forms of heparin were most effective in blocking virus infection.

Owing to their inherent structural features, certain sulfated glycans can exert therapeutic effects against infections caused by pathogenic microorganisms. A study by Pomin *et al.*, showed that administering sulfated glycans can disrupt the pathogen protein-host glycosaminoglycan (GAG) complex formation causing impairment of microbial binding onto host cells [55]. Similarly, sulfated GAG, glycosphingolipids and lectins have been shown to inhibit DENV entry [56]. Heparan sulfate mimics, such as suramin, pentosan polysulfate, and PI-88, SPGG [57,58] have been reported to be effective against multiple viruses including herpesviruses [4,59,60]. The inhibitory activity of HS mimics, including these compounds, is believed to be due to their association with GAG binding sites of the putative receptor-binding domain on the viral protein [4,61]. Thus, HS mimics can inhibit virus adsorption and entry.

Overall, the data from these studies indicate that *dp* of GAGs as well as specific sulfation patterns govern CMV infection of cells. These studies show the promise of highly polymerized sulfated-HS as effective anti-CMV agents. Future studies will be aimed at confirming the CMV glycoproteins that specifically bind to HS on cell surface and their possible structural illustrations.

## Supporting information

**S1 Fig. Bio-Gel P10 size exclusion column chromatogram of enoxaparin separation.** Fractions were collected and UV readings at 232 nm were taken for each fraction to reconstruct the chromatogram. Samples were pooled to obtain the oligosaccharide fractions of the desired size.
(TIF)

**S2 Fig. ESI-MS of 6-*O*-desulfated Arixtra.** The most abundant MS masses were consistent with the loss of the three 6-*O*-sulfates from Arixtra.
(TIF)

**S3 Fig. MS/MS analysis of the -4 charge state of Arixtra-3SO$_3$.** Glycosidic bond cleavages isolate desulfation to one desulfation event in the two non-reducing end residues; one desulfation event in the two reducing end residues, and one desulfation event in the central GlcNS. This pattern is consistent with 6-*O*-desulfation.
(TIF)

**S4 Fig. Effect of GAG treatment on cell viability of HFF cells.** Primary HFF were pretreated for one hour with 10 μM of 1) 6-*O*-desulfated Arixtra, 2) Regular Arixtra, 3) Heparin sodium salt from porcine intestinal mucosa (PIHSS), 4) Enoxaparin, or series of heparin oligosaccharide from enoxaparin: 5) *dp2*, 6) *dp4*, 7) *dp6*, 8) *dp8*, 9) *dp10*, 10) *dp12*, 11) *dp14*, 12) *dp16*, 13) *dp18*, 14) *dp20*, 15) > *dp20* or control (dH$_2$O). Cells were either mock infected **(A)** or infected with HCMV (Towne strain) virus at an MOI of 3.0 **(B)** in the presence of test glycosaminoglycans. Cells were harvested at 5 days post-infection and cell viability was assessed using Trypan Blue exclusion assay. Results are representative of three independent replicates. Standard error of mean was plotted as error bars.
(TIF)

**S5 Fig. Expression and analysis of HCMV glycoprotein B (gB).** HCMV gB was expressed and purified for GAG binding assays as described in Materials and Methods. Protein purification was verified by SDS-PAGE followed by staining with Coomassie Brilliant Blue. **(A)** Coomassie protein gel of purified CMV gB as well as flow thorough and washes. **(B)** CMV gB protein gel lane analysis was performed by using Bio-Rad Image Lab Software. Coomassie purity of gB was determined to be 100% and molecular weight was 107.59 kD. Lane 1—protein ladder (Bio-Rad Precision Plus Protein Dual Color Standards), Lane 2—CMV gB column flow through, Lane 3—1$^{st}$ wash, Lane 4—2$^{nd}$ wash, Lane 5—blank, Lane 6—CMV gB.
(TIF)

## Acknowledgments

We would like to thank Jian Zhang and Jessica Kelly at Z Biotech, LLC (Aurora, Colorado, USA) for help with glycoarray experimental design and data analysis.

## Author Contributions

**Conceptualization:** Dipanwita Mitra, Mohammad H. Hasan, Robert J. Linhardt, Joshua S. Sharp, Lianchun Wang, Ritesh Tandon.

**Formal analysis:** Joshua S. Sharp, Ritesh Tandon.

**Funding acquisition:** Robert J. Linhardt, Joshua S. Sharp, Lianchun Wang, Ritesh Tandon.

**Investigation:** Dipanwita Mitra, Mohammad H. Hasan, John T. Bates, Michael A. Bierdeman, Dallas R. Ederer, Rinkuben C. Parmar, Lauren A. Fassero, Quntao Liang, Hong Qiu, Fuming Zhang, Robert J. Linhardt, Joshua S. Sharp, Lianchun Wang, Ritesh Tandon.

**Methodology:** Dipanwita Mitra, Mohammad H. Hasan, John T. Bates, Michael A. Bierdeman, Dallas R. Ederer, Rinkuben C. Parmar, Lauren A. Fassero, Quntao Liang, Hong Qiu, Vaibhav Tiwari, Fuming Zhang, Robert J. Linhardt, Joshua S. Sharp, Lianchun Wang, Ritesh Tandon.

**Project administration:** Robert J. Linhardt, Joshua S. Sharp, Lianchun Wang, Ritesh Tandon.

**Resources:** John T. Bates, Michael A. Bierdeman, Quntao Liang, Vaibhav Tiwari, Fuming Zhang, Robert J. Linhardt, Joshua S. Sharp, Lianchun Wang, Ritesh Tandon.

**Supervision:** Robert J. Linhardt, Joshua S. Sharp, Lianchun Wang, Ritesh Tandon.

**Validation:** Joshua S. Sharp, Lianchun Wang, Ritesh Tandon.

**Visualization:** Joshua S. Sharp, Ritesh Tandon.

**Writing – original draft:** Dipanwita Mitra, Mohammad H. Hasan, Joshua S. Sharp, Ritesh Tandon.

**Writing – review & editing:** Dipanwita Mitra, Mohammad H. Hasan, Robert J. Linhardt, Joshua S. Sharp, Lianchun Wang, Ritesh Tandon.

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
