## [Decision Letter · Decision Letter 0]

24 Apr 2021

Dear Dr. Tandon,

Thank you very much for submitting your manuscript "The Degree of Polymerization and Sulfation Patterns in Heparan Sulfate are Critical Determinants of Cytomegalovirus Entry into Host Cells" for consideration at PLOS Pathogens. As with all papers reviewed by the journal, your manuscript was reviewed by members of the editorial board and by several independent reviewers. 

We sent the manuscript to three experts with expertise in herpesviral attachment and entry. All reviewers were in agreement that while very interesting, there were major concerns about the work and some of the data are self-contradictory. Additionally, some of the reviewers were not convinced about the novely of the experiments. The reviewers' comments are listed below and we hope you will find them helpful. If you are able to answer these concerns, we will be happy to consider a revised manuscript. Please note that the revised manuscript will be sent back to the reviewers for re-evaluation.

We cannot make any decision about publication until we have seen the revised manuscript and your response to the reviewers' comments. 

Sincerely,

Eain Murphy

Associate Editor

PLOS Pathogens

Blossom Damania

Section Editor

PLOS Pathogens

Kasturi Haldar

Editor-in-Chief

PLOS Pathogens

orcid.org/0000-0001-5065-158X

Michael Malim

Editor-in-Chief

PLOS Pathogens

orcid.org/0000-0002-7699-2064

Reviewer's Responses to Questions

**Part I - Summary**

Reviewer #1: This manuscript uses novel glyco arrays, peptides, and sulfation enzyme knockout lines in an attempt to map whether the degree of sugar polymerization and/or sulfation is important for MCMV or HCMV entry.

Reviewer #2: The authors have tried to demonstrate that structural specificities including sulfation patterns within heparin/Heparan sulfate play important roles in CMV entry and infectivity. While the area of research may be significant, many additional experiments are needed to improve novelty and reduce confusion and self contradiction.

Reviewer #3: The manuscript by Hasan et al addresses the role of specific cellular glycan modifications to infection by human cytomegalovirus. Glycan modification of mammalian cell surface proteins are complex with a tremendous variation in the lengths of the polymer chains and the chemical linkages between the different carbohydrate moieties. Many viruses use glycan modifications as receptors and the various branched structures can have major impacts on tropism; specific types of structures being characteristic of specific tissues or even mammalian species. The general view is that herpesviruses use heparin sulfate glycan modifications for initial adsorption, which then facilitates the binding to specific entry receptors (e.g., nectin for HSV). Here, Hasan and colleagues characterize the use of HS for cytomegalovirus infection, focusing on the polymer length and the sulfate linkage. They use an array approach to measure the binding of CMV to a variety of synthetic glycosaminoglycans and hone in on specific heparan sulfate characteristics for further study. Further experiments showing that soluble HS can interfere with CMV infection, and experiments using cell lines deficient in the enzymes that generate specific sulfate linkages lead the authors to conclude that the CMV bind principally to HS comprised of longer polymer chains bearing a 3-O-S sulfate linkage. Authors’ then speculate on the significance of this conclusion suggesting potential use in development of antiviral therapeutics.

A number of concerns outlined below limit enthusiasm for the study. In summary: 1) Experimental rigor is low given the lack of appropriate and rigorous statistical analyses of significances (especially in cases where the measured differences are small.) Moreover, some experimental protocols do not seem well matched to the hypotheses and this creates valid, alternative interpretations (see below for more detail). Together these concerns cast doubt on the validity of the conclusions. 2) The studies were conducted with only the “Towne” strain of HCMV. Given what is now appreciated about the genetic diversity of HCMV in vivo, including additional strains would increase the impact of the study. As a related point, there was not effort to identify what HCMV glycoprotein(s) are involved in the proposed interactions. Prior research has implicated gB and gM/gN in binding to GAGs, but in neither case has this been extensively studied. Both glycoprotein complexes exhibit some degree of genetic diversity. 3) A better discussion of the benefits and drawbacks of developing therapeutics targeting this stage of HCMV infection would increase the impact of the study. - Brent Ryckman (University of Montana)

**Part II – Major Issues: Key Experiments Required for Acceptance**

Reviewer #1: 1) needs to have the full panel of mouse GAG modification knockout lines tested.

2) Characterization/quantification of the purity of their different fractionation experiments

3) Inhibition analysis with different tropic strains of HCMV

Reviewer #2: Fig. 1 is unnecessary. Similar or identical figures exist in many papers including at least one by the authors here (PMID: 22773448). Please just refer to the published paper(s).

Figs.2/3, this data will be more meaningful (informative) if binding to gB was also measured in parallel and compared with the data shown here. This would also allow the authors to measure the binding affinity more precisely, something that has been hard to achieve in the field. Also, how do heparin structures (sulfation patterns etc.) compare with HS structures present in the glycoarray shown in Fig. 3?

Fig. 3, instead of Fig.1 the common heparin and HS structures (disaccharide or higher) that contribute to binding should be shown.

Fig. 4, the results are confusing and correlation with heparin/HS structures is not well described. Higher polymerization means higher negative charge availability to bind positive residues on gB or viral envelope, nothing specific or truly novel here. If the point is to show that heparin blocks viral growth then that point was already shown by Teresa Compton and colleagues almost three decades ago (PMID: 83847).

Figs. 5/6, argument similar to above (#4) applies here. Also, what is the new message here? Basically, multiple repetitions of results from Fig. 4. Not very impressive inhibition of infection. No statistics is performed to determine the significance.

Figs. 8/9, the results do not compare well. While the virus replicates the worst in 3OST1 KO cells (Fig. 8) it shows the best entry in the same cells (Fig. 9). Contradictory results, what is going on? Also, why does the double knockout support better replication (Fig. 8 statistical analysis suggests that results are significant)? Structural analysis of the GAGs from the double KO should be made and compared with single KOs.

Fig. 10, what is the purpose or novelty of this experiment? Similar use of anti HS peptides has been demonstrated against CMV in multiple papers by Tim Sparer’s group.

Reviewer #3: Suggested new experiments to strengthen the study.

• Some indication of that HCMV glycoprotein(s) are involved in the interactions proposed.

• I don’t understand why the experiments in Figs 4-5 were done by pre-incubating the cells with the GAG derivatives. If the hypothesis is that the virus binds to these GAG structures on the cells, then would it not make more sense to pre-incubate the virus with the GAGs to “block” the sites on the virus before you add the virus to the cells (Like using soluble receptor). This would also lessen the likelihood of effects on the cell physiology.

• Showing “dose responses” in experiments such as those in Figs 4-5 (potentially others) would also help strengthen the conclusion of preferential binding to specific GAGs.

**Part III – Minor Issues: Editorial and Data Presentation Modifications**

Reviewer #1: Really arrays in Fig. 2/3 could be combined into 1 figure A and B. Figure 3 is a just a more detailed analysis of the HS array in figure 2.

Previous publications do not indicate that CS or DS affect HCMV binding. Why does this array disagree with this? Length of the GAGs or sulfation? How much do these arrays relate to what is seen on actual susceptible cells?

Fig. 5 the degree of reduction is only 1-1.5 log reduction in titers at the highest dp. Although statistically significant, it also means that ~90% of the virus is NOT dependent on the dp and or other entry mechanisms.

Fig. 7 could be moved to supplemental data as this would not be expected to cause cell death. Important but not necessary in the body of the paper.

Fig. 8 These MCMV cell lines only show a modest reduction implying multiple HS/modifications are important for infection. There are a whole panel of the these lines available from the Wang /Esko lab in order to provide a more complete picture of the important of the different sulfations on viral entry. Their paper from 2018 is pretty comprehensive as they knocked out a many of the enzymes involved in the HS/sulfation pathway. There needs to be an explanation for why the double KO does not lead to a “cumulative” (as mentioned in the discussion).

The switching and in complete analysis between HCMV and MCMV is confusing. Complete the analysis in either one or the other. Much of the previous works has been completed for HCMV so a complete analysis of MCMV entry would be more novel.

An interesting concept that could make the manuscript more complete/interesting would be the inhibition experiments with endothelial, fibroblast, macrophage-tropic viruses and how they are or are not interfered with the dp HS/sulfations.

Fig. 9 This is unconvincing and includes mutant lines not described/used in Fig. 8. There are potentially interesting cell line mutants that could provide interesting information if carried out as in Fig. 8’s set up. It would be better to use a GFP expressing virus to address entry instead of these western blots.

Fig. 10 The number of foci/well are too low. Usually (for consistency) you want closer to 50 /well. There are many different types of HS binding peptides. There were from Tiwari et al but there are others out there that block entry more generally and would be good to use as a positive/negative control. They would need to show the specificity of theses peptides on the cell types used. Why not see if they block entry on the different mouse KO lines? Would it further decrease entry/infection?

Should be using STD not SEM for figures. Also should show each data point to help in the interpretation.

Reviewer #2: (No Response)

Reviewer #3: • Line 70. Didn’t Hetzenecker 2015 show that HCMV can also use endocytosis pathways in fibroblasts?

• Fig2. Is there some kind of statistical error/significance test that can be used? The legend indicates six replicates, which implies that the values plotted are averages, is this the case? Fig 3 is the same type of analysis and has error bars, although the legend does not state what they represent. In any case, some analysis beyond standard deviation or standard error would likely be called for here.

• Materials and methods indicates that HCMV Towne strain was used. Could the authors provide more information. Is this BAC derived? Or the ATCC Towne? (which is a mix of genotypes, right? Long and short at least?). Or is this some other variant of Towne? Moreover, given the explosion of research relating to HCMV genetic diversity, how can the authors conclude that this heparin interaction is characteristic of all strains or variants of HCMV? Particularly since there is no indication of what CMV glycoproteins are binding these GAGs (assuming this conclusion is true: see other comments).

• Line 249: I understand the point of this control, but if it is a negative control, then why name it a positive control only to have to explain here?

• Fig 4. It is unclear from the results text or the legend if the GAG derivatives were present throughout the 5 day experiment. Maybe it is in the materials and methods, but it is important enough to be stated in the legend. Is this testing initial infection (entry) or also subsequent spread over 5 days? Again, some statistical analysis beyond standard error would be good, especially since the data are normalized to %. What was the dynamic range? (how many foci were counted in the control wells?

• Fig 5 seems to be the same experiment as in Fig 4, but with an alternative readout (viral titer recovered at 5 days vs # of GFP+ foci at 5 days). What is the point of the plot in 5B? Is seems to be only a subset of the exact data from 5A, just plotted as dots connected by a line rather than as columns. If you are trying to show a correlation with the degree of polymerization, there are specific correlation tests that may be appropriate. The more interesting comparison would seem to be between the two readouts; ie, why does the titer recovered at 5 days not track with the GFP readout? If the GAG derivatives are present for the entire experiment, are the progeny release to the supernatant affected? Does this effect then carry over to the plaque assay titration? Or are the cells themselves affected by the GAGs?

• Fig 7A. Vital dye exclusion is a very coarse assessment of potential effects of the GAGs on the cells. Is it possible that the physiology of the cell is altered in a way that does not “kill” the cell?

• Line 294/Fig7B. Statement in results does not seem to match data shown.

• Fig 8. Legend states that the error bars represent standard error, but there is no indication of the statistical tool used for the “asterisks” of significance. If these asterisks are only indicating which error bars do not overlap with the control, they should be removed and an appropriate test used. (Probably one-way ANOVA+Dunnetts post hoc). Some of the differences seem quite small.

• Fig 9. It would be much easier to interpret these data if the lane order in the blots matched the order of the relative density columns left to right.

• Lines 315-318. Is it really fair to say that the data in Fig 9 indicate differences in “entry?” I grant you that the notion suggested above that the GAG derivatives might affect cell physiology is a bit of the “formal caveat” (meaning it is formally possible, but maybe not all that likely). But here in fig 9, the cells have been asked to grow without the normal GAGs modification pathways. This, by definition, means their physiology is altered, no? I’m no expert on glycan modifications, but seems to me that the pathways are critical to the cell being able to move membrane proteins from the ER to the plasma membrane and then recycle them. It’s hard to imaging these cells are not physiologically abnormal. Maybe they just can’t support IE expression very well. Even if the cells are able to support IE expression normally after entry, how would you know that the entry defect is due to the lack of binding to the GAGs and not that the altered glycan processing pathways impacted the expression of specific receptors (if this was an HCMV experiment, I would suggest PDGFR-alpha, for example)?

• Line322. Is there some citation or supplemental data to support the specificity of these peptides as stated?

• Fig 10. As suggested above, is it possible that these peptides alter cell physiology? In fact, this seems likely in this experiment since they are designed to actually bind to the cells, and may well engage signaling pathways similar to the natural ligands for the GAGs as indicated by the authors in Lines 42-43 (cytokines; growth factors etc).

PLOS authors have the option to publish the peer review history of their article (what does this mean?). If published, this will include your full peer review and any attached files.

Reviewer #1: No

Reviewer #2: No

Reviewer #3: **Yes: **Brent Ryckman
---

## [Editor Report · Decision Letter 1]

15 Jul 2021

Dear Dr. Tandon,

We are pleased to inform you that your manuscript 'The Degree of Polymerization and Sulfation Patterns in Heparan Sulfate are Critical Determinants of Cytomegalovirus Entry into Host Cells' has been provisionally accepted for publication in PLOS Pathogens.

Best regards,

Eain A Murphy, Ph.D.

Associate Editor

PLOS Pathogens

Blossom Damania

Section Editor

PLOS Pathogens

Kasturi Haldar

Editor-in-Chief

PLOS Pathogens

orcid.org/0000-0001-5065-158X

Michael Malim

Editor-in-Chief

PLOS Pathogens

orcid.org/0000-0002-7699-2064

Dear Dr. Tandon,

We, the editors, have reviewed your resubmission and have come to the conclusion to render a decision of Accept for this version as it is apparent that you have taken the original reviewers' concerns seriously, have made significant modifications to the manuscript and have vastly improved the work in doing so.

Congratulations to you and your team.

Cheers,

Eain Murphy
---

## [Editor Report · Acceptance letter]

3 Aug 2021

Dear Dr. Tandon,

We are delighted to inform you that your manuscript, "The Degree of Polymerization and Sulfation Patterns in Heparan Sulfate are Critical Determinants of Cytomegalovirus Entry into Host Cells," has been formally accepted for publication in PLOS Pathogens.

Best regards,

Kasturi Haldar

Editor-in-Chief

PLOS Pathogens

orcid.org/0000-0001-5065-158X

Michael Malim

Editor-in-Chief

PLOS Pathogens

orcid.org/0000-0002-7699-2064